Effects of low-level electric current on the growth of Amphistegina lobifera and its photosynthetic diatom endosymbionts

Rebecchi Federica federica.rebecchi@uniurb.it 1 2
Lattanzi Davide 3
Abramovich Sigal 4
Ambrogini Patrizia 3
Frontalini Fabrizio 1
Schmidt Christiane 2 5
1 Department of Pure and Applied Sciences, University of Urbino , Urbino , Italy
2 Leibniz Centre for Tropical Marine Research (ZMT) , Bremen , Germany
3 Department of Biomolecular Science, University of Urbino , Urbino , Italy
4 Department of Earth and Environmental Sciences, Ben-Gurion University of the Negev , Beer Sheva , Israel
5 Helmholtz Centre for Geosciences (GFZ) , Potsdam , Germany
Banaszak Anastazia
Electronic publication date: 2025 Oct 20
Publication date: 2025
Volume: 13
Electronic Location ID: e20160
Received 2025 Mar 20; Accepted 2025 Sep 10
Copyright: ©2025 Rebecchi et al.
Copyright year: 2025
Copyright holder: Rebecchi et al.
License: This is an open access article distributed under the terms of the Creative Commons Attribution License, which permits unrestricted use, distribution, reproduction and adaptation in any medium and for any purpose provided that it is properly attributed. For attribution, the original author(s), title, publication source (PeerJ) and either DOI or URL of the article must be cited.
License URL: https://creativecommons.org/licenses/by/4.0/

Keywords: Marine biotechnology, Electrolytic mineral accretion, Electric stimulation, Low-cost prototype device, Calcification, Benthic foraminifera

Funding: the DFG (German Research Foundation) grant SYMBIO-AID No. 444059848 PON REACT-EU program and ReMeSt PhD school of the University of Urbino Carlo Bo, Italy DOT197EXS3-7 DO The experimental part of this study was funded through the DFG (German Research Foundation) grant SYMBIO-AID, which was an Individual Research Grant given to Christiane Schmidt (No. 444059848). Federica Rebecchi received a stipend to conduct a research stay in Bremen from PON REACT-EU program and ReMeSt PhD school of the University of Urbino Carlo Bo, Italy (grant number DOT197EXS3-7 DO). The funders had no role in study design, data collection and analysis, decision to publish, or preparation of the manuscript.

==============================
Larger benthic foraminifera (LBF) are key carbonate producers and significantly contribute to carbonate reef sediments. As the ongoing climate change threatens the calcification capacity of many marine organisms, novel approaches are being explored to support reef resilience. Among these, low-voltage electric stimulation has shown promise in enhancing calcification in corals and other marine calcifiers by stimulating electrodeposition of calcium carbonate. However, the potential of this technique to support calcification in LBF has not yet been assessed. To close this gap, the present study investigates the effects of low electric current densities on the LBF species Amphistegina lobifera. To avoid inducing mortality, the current densities were carefully selected based on previous findings and were applied in two 30-day experiments. Pulse-Amplitude Modulation (PAM) fluorometry (Fv:Fm) was used to measure the photosynthetic efficiency of the diatom endosymbionts, and total pigment content (Chl a) was analysed via a plate reader to assess pigment changes due to the electric stimulation. Growth was analysed by measuring the maximum diameter and counting the formation of new chambers labelled with the fluorogenic dye calcein. The results of both experiments show that electric stimulation did not affect the maximum quantum yield (Fv:Fm) and Chl a content. Furthermore, all treatments exhibited positive growth, but no significant growth enhancement was observed compared to the controls. The highest growth and chamber formation rate were found at current densities of 1 and 1.43 µA/cm2, which represent the highest growth rates obtained in the experiments, as an additional increase in current density to 2.86 µA/cm2 did not seem to further enhance growth. These results suggest that low electric current can influence foraminiferal growth, and the conditions necessary for a significant enhancement remain to be investigated.

Introduction

Increased CO2 emissions due to global climate change have lowered ocean pH and altered carbonate chemistry (Caldeira & Wickett, 2003; DeVries, 2022). The reduction of carbonate ions essential for marine calcifiers to build their skeletal structures adversely affects their growth, reproduction, and survival rates (Jiang et al., 2023; Kroeker et al., 2013; Orr et al., 2005). However, the response to elevated CO2 and dissolved inorganic carbon (DIC) is highly variable among calcifying organisms, with some species showing reduced calcification and others remaining unaffected, highlighting species-specific responses depending on life stage and type of calcium carbonate (Ries, Cohen & McCorkle, 2009; Kroeker et al., 2010). In addition, the ability to utilize different forms of inorganic carbon may enhance resilience to elevated CO2 (De Goeyse et al., 2021).

To tackle pressures on coral reefs and to preserve their ecosystem integrity not only for its biological functions but also tourists, several approaches of marine bioremediation and coral reef restoration have been proposed (Fragoso ados Santos et al., 2015; Hughes et al., 2023; Burdett et al., 2024). Among such innovative biotechnological approaches, the application of low electrical fields generated by the Biorock electrolytic method has been shown to significantly enhance the settlement, growth, and survival of marine organisms under different stress conditions (Goreau, 2022; Goreau & Hilbertz, 2005). The Biorock method, first introduced by Hilbertz (1979) and then developed for the growth of aquatic organisms by Hilbertz & Goreau (1996), is an electrochemical technique that utilizes low-voltage electrical charges to create an electrical field, concentrating dissolved mineral ions around the electrodes and accelerating the calcification rate (Hilbertz, 1992; Sabater & Yap, 2004). Calcifying organisms rely on Ca2+ and HCO3− ions for skeleton construction (Allemand et al., 2004). These ions precipitate as calcium carbonate when supersaturation is reached (Barron et al., 2018). This process depends on ion concentrations in seawater and the pH within the calcification centre (Comeau et al., 2017). The electrochemical techniques utilize electric fields to enhance natural calcification processes. Applying a low voltage and direct electrical current across a cathode and an anode, seawater undergoes electrolysis, producing hydrogen gas and hydroxide ions. This reaction increases the pH around the cathode, creating conditions that promote the precipitation of calcium carbonate (Hilbertz, 1992). Calcifying organisms like corals located near the cathode benefit from higher concentrations of mineral ions and an alkaline environment, which together accelerate their calcification and growth (Goreau, 2014). This biotechnological approach has already found application in marine ecosystem restoration to increase coral growth and survival (Goreau & Hilbertz, 2005; Goreau & Trench RK, 2013). Additionally, it has been applied to increase oyster growth and stimulate the cultivation of pearl oysters (Berger et al., 2012; Karissa et al., 2012; Shorr et al., 2012).

In calcifying organisms, calcification and photosynthesis are commonly linked, with both processes dependent on carbonic anhydrases (CAs) activity. These enzymes, widespread across the animal kingdom, catalyse the interconversion of CO2 and HCO3−, providing inorganic carbon for both photosynthesis and calcification (Bertucci et al., 2013). Carbonic anhydrases comprise a large family of isoenzymes, and in corals, several isoforms have been identified, including cytosolic, mitochondrial, membrane-bound, and CA-related proteins, each involving in specific roles in calcification and symbiont photosynthesis (Moya et al., 2008; Bertucci et al., 2013). Photosynthesis reduces CO2 concentration locally, promoting the production of additional CO2 from bicarbonate, creating a microenvironment that enhances carbonate ion availability for calcification. The adenosine triphosphate (ATP) generated during photosynthesis also supports the active transport of ions, like Ca2+, to the calcification site, mediated by enzymes like Ca-ATPase (Al-Horani, Al-Moghrabi & De Beer, 2003). CA can also catalyse the reaction from CO2 to HCO3− and, ultimately, CO32−, which increases local carbonate ion availability and supports CaCO3 precipitation (Chen, Gagnon & Adkins, 2018).

Benthic foraminifera, single-celled marine organisms, play a significant role in the ocean’s calcium carbonate (CaCO3) cycle by secreting calcium carbonate tests (Lee & Anderson, 1991). Among these, photosymbiotic-bearing larger benthic foraminifera (LBF) are crucial carbonate producers in warm and euphotic habitats, contributing up to 80% of the global foraminiferal reef carbonate that represents roughly 5% of total reef carbonate production (Langer, Silk & Lipps, 1997; Langer, 2008a; Langer, 2008b). LBF are ideal model organisms for studying growth due to their periodic and visually detectable calcification process (Erez, 2003). Due to their abundance and small dimension, they are suitable candidates for testing electrochemical techniques. In addition, LBF share characteristics with reef-building corals, such as hosting and relying on a variety of endosymbionts (e.g., diatoms, dinoflagellates, red algae) (Lee et al., 2010) for growth and calcification (Hallock, 1999). However, contrary to corals, they associate with a more diverse pool of endosymbiotic algae, not only Symbiodinium (Baker, 2003; Rosic et al., 2015). The photosynthetic fixation of CO2 by their endosymbiotic algae increases the saturation state of carbonate, thereby enhancing the calcification rate in LBF (Wolf-Gladrow, Bijma & Zeebe, 1999; Glas et al., 2012). In benthic foraminifera, calcification is linked to proton pumping, which lowers the pH outside the protective envelope and thereby increases local pCO2. This creates a strong pCO2 gradient that drives CO2 diffusion toward the site of calcification (SOC) where elevated pH promotes the conversion of CO2 into carbonate ions, which combine with Ca2+ to precipitate CaCO3. The elevated pH at the SOC enhances the availability of carbonate ions for calcite precipitation, a mechanism catalysed by CA, which plays a critical role in facilitating the inward diffusion of CO2 and supporting the calcification process (Glas et al., 2012; Toyofuku et al., 2017). Elevated pH at the SOC has been documented in LBF species Amphistegina lobifera, where direct measurements revealed its association with active calcification (Rollion-Bard & Erez, 2010) as well as in various other calcifying organisms (Sutton et al., 2018). Similar to studies in corals, based on this physiological interplay, we hypothesize that electrical stimulation might enhance photosynthetic activity and, thereby, promote growth and calcification in LBF. It should be noted that potential seawater pH changes induced by the applied current were not assessed in this study, and our interpretations focus exclusively on the physiological effects of electrical stimulation.

To explore these physiological effects under controlled conditions, we selected Amphistegina, a prevalent genus of diatom symbiont-bearing foraminifera. This genus is widely distributed on coral reefs and tropical carbonate shelves worldwide and possesses a calcium carbonate shell (Hallock, 1999). Amphistegina is highly responsive to environmental changes, making it an important biological indicator (Hallock et al., 2003). Several studies have explored the impact of various environmental stressors on these organisms, including excess solar energy (Talge & Hallock, 2003; Williams & Hallock, 2004; Hallock et al., 2006; Prazeres, Martins & Bianchini, 2012), thermal stress (Schmidt et al., 2011; Stuhr et al., 2018; Titelboim et al., 2021b), ocean acidification (Prazeres, Uthicke & Pandolfi, 2015; Vogel & Uthicke, 2012), heavy metals (Titelboim et al., 2021a; Ciacci et al., 2022; Schmidt et al., 2025) and their effectiveness as bioindicators of water quality (Prazeres, Martins & Bianchini, 2012). Their biological traits, including their responsiveness to environmental stimuli and ease of monitoring growth processes, make Amphistegina a highly suitable candidate for testing novel electrochemical techniques in controlled experimental setups.

To date, the electrochemical technique to enhance the precipitation of dissolved ions in seawater and its consequent effect on the growth rate has not been applied to LBF. To provide a robust method with reduced mortality, a previous experiment on electrical current stimulation compared constant and pulsed types of direct current (Rebecchi et al., 2023). For this study, a prototype generating low direct current has been developed for electrochemical experiments and other applications (Lattanzi et al., 2024). The results revealed that direct pulsed current had a limited negative impact on the viability of the symbiont-bearing foraminiferal species Amphistegina lessonii compared to direct constant current. This experiment enabled the appropriate and harmless electrical current density threshold on LBF and demonstrated that direct pulsed current is the most suitable choice (Rebecchi et al., 2023).

To unravel if the electrochemical stimulation promotes the precipitation of calcium carbonate through electrolysis in seawater in LBFs as already revealed for other marine calcifying organisms and to document their growth rate for enhancing reef calcification, the present study aims to investigate the effects of different electric current densities over a 30-day period on the growth rate of the LBF Amphistegina lobifera and the physiological performance of their endosymbiotic diatoms. Specifically, we performed two experiments that assessed: firstly, the photosynthetic performance of LBF endosymbionts by measuring the photosynthetic efficiency (Fv:Fm) and evaluating changes in total Chl a pigments; secondly, the growth rate of LBF through morphometric analysis and the formation of new chambers by calcein labelling.

Materials & Methods

Sample collection

Living specimens of Amphistegina lobifera were collected at Shikmona (Lat. 32.82601, Long. 34.95577, in northern Israel, Mediterranean Sea) at 50 cm water depth in January 2023. Samples containing living foraminifera in natural seawater were express shipped to Leibniz Centre for Tropical Marine Research (ZMT) in Bremen (Germany). Prior to experiments, samples were acclimated for up to 3 weeks at 25 °C at 30-40 Photosynthetically Active Radiation (PAR) light intensities under a 12:12 light-dark cycle. Salinity was kept constant between 38–39, in artificial seawater (ASW: Tropic Marine® Classic, Germany) and replaced by 50% every seven days.

Experimental setup

Two experiments were carried out using the same low-pulsed electric current steps to determine 1. the effects of electric current on the physiology of its endosymbionts (using Pulse-Amplitude Modulation (PAM) Fluorometry) and 2. the growth rate of A. lobifera using more detailed morphological analysis using the calcein labelling and, in parallel, tracking the physiology of endosymbionts replicating the first experiment (Fig. 1). The design of each experiment was identical, except that in the second experiment, the calcein probe was added at the start of the procedure.

For each experiment, a total of 240 living specimens of A. lobifera were picked under an optical microscope (Stereomicroscope V8 Zeiss with Canon EOS 600D, magnification 1.2x) and randomly distributed in 15 culturing plates specifically designed to allow the electric stimuli to target LBF. A total of 16 specimens were used per plate and kept in 150 ml of ASW (Tropic Marine, Germany). Each culturing plate contained a silicon tube that supplied bubbles for optimal aeration of the samples. Additionally, electrodes were placed inside each culturing plate and connected to the electric stimulus generator. The experiment was carried out in a climate-controlled room (25 °C ± 3 °C) with a light level of 30-40 PAR under a 12:12 light-dark cycle. Salinity and temperature were also checked every week, and foraminifera were fed weekly with 150 µl of Nannochloropsis food mixture (Schmidt et al., 2015).

Figure 1 Experimental design overview.

(A) First experiment: effect of electrical stimulation on the ecophysiology of symbionts, including measurements of maximum quantum yield (Fv:Fm) using PAM fluorometry and Chlorophyll a (Chl a) content analyzed with a plate reader. (B) Second experiment: effect of electrical stimulation on ecophysiology of holobiont including morphometric analysis and the count of newly formed chambers using a stereomicroscope, and on symbiont physiology, including maximum quantum yield (Fv:Fm) measured with PAM fluorometry. Created in BioRender. Rebecchi et al. (2025) https://BioRender.com/91d0r5r.

Custom-made setup of electrical stimulation

The setup followed the design of a previous experiment conducted at Urbino University (Italy), where the viability of the genus Amphistegina was measured after exposure to pulsed direct current values (Rebecchi et al., 2023). However, in our experiment, we modified the design by using larger culturing plates to provide more space for each specimen and different electrodes, specifically steel electrodes, to achieve a better-distributed electric field. Based on the previous findings, we selected different low-pulsed current values: (A) 2.9 µA (corresponding to 0.29 µA/cm2); (B) 10 µA (corresponding to 1 µA/cm2); (C) 14.3 µA (corresponding to 1.43 µA/cm2); and (D) 28.6 µA (corresponding to 2.86 µA/cm2). These current densities were chosen as non-lethal, showing the highest percentages of living individuals, and not producing adverse physiological stress but sufficiently high to potentially stimulate growth. Each current treatment was replicated three times to ensure a well-replicated data set, as well as a control treatment, which was unpowered and exposed the foraminifera in the same conditions except for the electrical current. The experiment was conducted for a 30-day period, where ASW was replaced with fresh ASW every week by exchanging 30% of the total water volume.

Pulse-amplitude modulated fluorometry

The electrical current effects on the photosynthetic efficiency of the endosymbiotic diatoms were measured as maximum quantum yield (Fv:Fm) with the Imaging Pulse-Amplitude Modulated (PAM) fluorometry (Unit IMAG-CM, connected to a MAXI Head with a 300 W LED-array) developed by Walz (Schreiber, Schliwa & Bilger, 1986). The Imaging-PAM applies a pulse-modulated excitation and actinic light for estimation of chlorophyll fluorescence yield allowing homogenous measurements of the samples. The initial fluorescence (F0) was measured by applying a weak pulsed red light, followed by a saturation pulse of actinic light measuring the maximum fluorescence (Fm). The maximum quantum yield (Fv:Fm) was calculated as ((Fm-F0): Fm) automatically by the software (WALZ, Germany). For measurements, foraminiferal specimens were removed from the custom-made experimental setup and measured into the treatment water inside small petri dishes (volume 3–5 mL). In these dishes, the samples were at least dark-adapted for 15 min prior to measurements in complete darkness.

Chlorophyll analysis

The physiological responses of the algal symbionts were quantified by determining a change in the total pigments of the symbionts. The analysis for the total Chlorophyll a (Chl a) content was performed using the protocol of Schmidt et al. (2011). For each treatment, three specimens were pooled together for the analysis of Chl a. The wet weight of each foraminiferal specimen was measured in pre-weighed 1.5 mL Eppendorf vials, with an accuracy of 0.01 mg. All the samples were maintained under low light conditions and kept on ice throughout the analysis.

Growth measurements

The potential effects of electric current stimulation on the growth rate of LBF were measured on all individuals that were vital after the second experiment. The foraminiferal growth rate was calculated by capturing light microscope images (stereomicroscope V8 Zeiss with Canon EOS 600D, magnification 1.2x) before and after the experiment and analysing them using ImageJ software (64-bit Java, 1.54f; Schneider, Rasband & Eliceiri, 2012). The maximum diameter of each foraminifer was measured, and growth rates were calculated using the following formula (1). (1) finalmeasurementofmax.diameter−initialmeasurementofmax.diameterinitialmeasurementofmax.diameter∗100.

Statistical analyses were performed on the average growth per plate (i.e., replicate), considering only individuals that were alive and exhibited healthy coloration at the end of the experiment. Negative growth values were excluded to avoid potential bias from non-viable specimens. One specimen, from the plate stimulated at 2.86 µA/cm2, was accidentally lost during handling and could not be measured.

In addition, the growth rate was assessed by counting newly formed chambers during the experiment. This was achieved by using the green calcein probe, a fluorescent compound that binds with calcium in biomineralized structures allowing for the labelling and identification of newly formed chambers (Bernhard et al., 2004). According to Bernhard et al. (2004), incubation with calcein at a concentration of 10 mg/L does not negatively affect foraminiferal calcification or survival. However, Li et al. (2020) recommended an incubation using a lower calcein concentration (∼5 mg/L), as higher concentrations (10 mg/L) and incubation periods of several weeks (e.g., 6 weeks) were found to have negative effects on the foraminiferal community. Hence, we chose a lower dose (5 mg/L) to not additionally stress foraminifera or their symbionts. At the beginning of the experiments, the calcein solution was added to experimental containers (concentration of 5 mg/L calcein dissolved into ASW). The solution of calcein spiked ASW changed weekly. The counting of newly formed chamber was performed using an epifluorescence stereomicroscope (Zeiss Discovery V8) at the University of Urbino (Italy).

Statistical analysis

Differences of PAM, Chl a, and growth (i.e., dimension and newly formed chamber) among experimental conditions (i.e., control and treatments) were statistically analysed using a nonparametric Kruskal–Wallis H test. Kruskal–Wallis test was performed using the statistical software JMP® (Version 17, SAS Institute Inc., Cary, NC).

Results

Pulse-amplitude modulated fluorometry and chlorophyll a

The photosynthetic activity of Amphistegina endosymbionts was assessed through PAM fluorometry by measuring the dark-adapted yield (maximum quantum yield; Fv:Fm). The Fv:Fm values ranged from 0.58 to 0.62 over 30 days (Fig. 2A). The results showed that the mean maximum quantum yield remained mostly stable across all current densities in both experiments (Fig. 2A). The lowest values of maximum quantum yield were recorded at the highest current density (i.e., 1.43 µA/cm2 in the first experiment and 2.86 µA/cm2 in both experiments) (Fig. 2A). A Kruskal-Wallis test revealed no significant differences among treatments in both the first and second experiments.

Figure 2 Photosynthetic Performance and Chlorophyll a content in Amphistegina lobifera.

(A) Mean maximum quantum yield (Fv:Fm) measured during the first experiment (blue bars) and the second experiment (green bars); (B) mean Chlorophyll a content (μg/mg). Error bars represent standard deviation.

The Chl a content was measured to assess changes in the total biomass of the diatom endosymbionts. The Chl a content in the control was 68.4 ± 18.8 µg/mg and ranged from 57.7 ± 12.8 µg/mg at 1 µA/cm2 to 107.2 ± 33.9 µg/mg at the highest current density of 2.86 µA/cm2 (Fig. 2B). No significant difference between the treatments in Chl a content was found (Table 1).

Growth measurements

The growth rate (% maximum diameter increase) was measured for all specimens of A. lobifera exposed to different current densities in the second experiment.

The results showed a higher growth rate in specimens stimulated with a current density of 1 and 1.43 µA/cm2 (Fig. 3A). However, there were no significant differences in growth rates between the various current densities at the end of the experiment (Table 1). The mean growth rate among experimental conditions revealed a growth of 3.2%, 2.9%, 3.4%, 3.5% and 1.8% for control, 0.29, 1, 1.43 and 2.86 µA/cm2, respectively (Table 2). Interestingly, the highest percentages of mean growth were found for 1 and 1.43 µA/cm2, whereas the lowest percentages were associated with 2.86 µA/cm2 (Table 2).

Additionally, a higher percentage of newly formed chambers was labelled with calcein at the current densities of 1 and 1.43 µA/cm2, compared to the control (Fig. 3B). The mean number of newly formed chambers per individual varied across treatments, with 0.48 ± 0.24 chambers per individual at 1 µA/cm2 and 0.56 ± 0.11 chambers at 1.43 µA/cm2. In contrast, the mean values were similar between the control (0.32 ± 0.05) and the highest current density tested (2.86 µA/cm2), which resulted in a mean of 0.35 ± 0.19 chambers per individual (Fig. 3B). The percentages of newly formed chambers compared to control were at the highest (i.e., 50.1% and 76.3%) at 1 and 1.43 µA/cm2 (Table 2), while the lowest growth percentage (11%) compared to control was found for 2.86 µA/cm2. This suggested a slightly higher rate of chamber formation at these current densities but with no significant differences (Table 1).

Table 1 Kruskal–Wallis test for the maximum quantum yield (Fv:Fm), total Chlorophyll a content (Chl a in μg/mg), growth rate (%), and the newly formed chambers (%) among experimental control and treatment.

Results of the Chi-Square tests.

		ChiSquare	df	p-value	
Maximum quantum yield (Fv:Fm)	Exp 1	2.9559	4	0.5652	
	Exp 2	2.4393	4	0.6555	
Total Chl a (μg/mg)		5.2667	4	0.2610	
Growth rate (%)		1.2667	4	0.8670	
Newly formed chambers (%)		4.2347	4	0.3752	

Figure 3 Growth and chamber formation in Amphistegina lobifera.

(A) Growth rates expressed as the percentage of individual growth, and (B) the number of newly formed chambers labelled with calcein in Amphistegina lobifera exposed to different current densities. Error bars represent standard deviation.

Table 2 Summary of growth parameter in Amphistegina lobifera exposed to different current densities. Mean initial and final dimension, mean growth, percentage (%) of growth, mean of newly formed chambers and percentage (%) of newly formed chambers to control.

	Current density (μA/cm2)	
	Control	0.29	1	1.43	2.86	
Mean initial dimension (μm)	1,493.4 ± 40.8	1,410.1 ± 99.5	1,448.7 ± 45.7	1,440.1 ± 66.7	1,443.8 ± 71.2	
Mean final dimension (μm)	1,541.1 ± 31.3	1,450.7 ± 80.3	1,497.8 ± 33.1	1,489.9 ± 70.5	1,470.3 ± 90.4	
Mean growth	47.6	40.6	49.1	49.8	26.4	
% of growth	3.2	2.9	3.4	3.5	1.8	
Mean newly formed chambers	0.32 ± 0.05	0.44 ± 0.23	0.48 ± 0.24	0.56 ± 0.11	0.35 ± 0.19	
% of newly formed chambers to control		37.1	50.1	76.3	11.0	

Discussion

Effects of low-level electric current on Amphistegina lobifera endosymbionts

The results of this study provide compelling evidence that Amphistegina lobifera can tolerate a range of electric current densities (0.29–2.86 µA/cm2) without any measurable adverse effects on their ecophysiology measured as photosynthetic activity (maximum quantum yield, Fv:Fm), Chl a content, and growth rates.

The Fv:Fm decreases at the highest applied currents (i.e., 1.43 and 2.86 µA/cm2). Despite these variations, values remain mostly stable between 0.5–0.6, consistent with maximum levels measured in healthy photosystems (Schmidt et al., 2011; Vogel & Uthicke, 2012). The increase in the maximum quantum yield (Fv:Fm) is also in line with the increase in growth during the second experiment. We can therefore infer that electric current does not negatively impact the diatom symbionts, as also evidenced by the slight increase of Chl a content at 1.43 and 2.86 µA/cm2 and the holobiont performance. In contrast to our findings, a study on coral endosymbionts reported that electric current stimulation increased the maximum quantum yield, resulting in enhanced coral growth (Huang et al., 2020). A possible explanation for this discrepancy could be the differences between the endosymbiotic species hosted by LBF and corals, as corals associate with Symbiodiniaceae symbionts and the LBF species used for this study harbors endosymbiotic diatoms. The concentration of those symbionts was measured to further shed light on stimulative effect of the electrical currents on the symbiont-host relationship.

The Chl a content, an indicator of symbiont biomass, shows no significant differences across the tested current densities, except for a reduction at one µA/cm2 and, an overall increase at 1.43 and 2.86 µA/cm2. The substantial stability suggests that the density (quantity) of symbionts reflected as total Chl a remains mostly unaffected by the electric current stimulation. In corals, Chl a content commonly correlates with symbiont density, and studies have shown that electrical stimulation can increase symbiont densities while reducing Chl a content per cell, potentially due to the increased energy availability provided by the electrical field (Goreau, 2014). However, in benthic foraminifera, as reported by Talge & Hallock (2003), symbiont density can be assessed through cytological observations. Without this approach, accurately counting symbionts becomes highly challenging, making it difficult to determine whether the observed reduction in Chl a at 1 µA/cm2 resulted from a decrease in symbiont numbers or a reduction in Chl a content per symbiont cell. These findings underscore the need for refined methods to quantify symbiont density and Chl a content more accurately, particularly in LBF, in order to better understand the relationship between electric stimulation and symbiont responses.

Effects of low-level electric current on Amphistegina lobifera growth

This study shows no significant enhancement of growth compared to control, all treatments exhibit positive growth, indicating a resilience to the applied electrical stimulation. However, we observe an increase (non-significant) in foraminiferal growth at a current density of 1 and 1.43 µA/cm2. The observed growth is further confirmed by the highest percentage of newly formed chambers at the current density of 1 and 1.43 µA/cm2, suggesting that moderate electric currents may support cellular processes that facilitate calcification or chamber formation, even if not significantly, compared to the controls. We initially hypothesised that the electric current would increase the foraminiferal growth by enhancing photosynthetic activity based on known links between photosynthesis, calcification, and carbonic anhydrase (CA) activity. Previous research has demonstrated that CA activity decreases with lower symbiont densities, as observed in symbiotic scyphozoans (Estes, Kempf & Henry, 2003). In corals, a reduction in CA activity similarly reduces photosynthetic rates due to carbon limitation (De Beer et al., 2000). In the case of LBF, A. lessonii, the inhibition of photosynthesis reduces calcification, suggesting that photosynthesis actively promotes calcification, potentially through CA’s role in increasing CO2 availability at the SOC (De Goeyse et al., 2021). Contrary to our hypothesis, the electric current has not stimulated, at least significantly, the photosynthetic activity. These findings suggest that while electric current may influence the growth of benthic foraminifera, its enhancement and deviation from standard growth conditions under current experimental conditions is limited. Further studies need to test nutrient enrichment, light and electric stimuli in combination to further increase the newly built calcite in a timely manner. Additionally, a newly developed protocol on inducing aposymbiosis in LBF through a rapid menthol-DCMU bleaching method has made it possible to study calcification and growth on LBF without symbionts and compare them to LBF with different symbiont types (Schmidt et al., 2025). These studies not only provide a deeper understanding of the calcification process but also shed light on the role of symbionts, thereby enriching our knowledge of the complex factors that influence foraminiferal growth.

The findings from this study align with broader research on the effects of electric current on other calcifying marine organisms. The use of electric current to promote growth has been widely investigated in corals and oysters, through the Biorock method, an electrochemical technique designed to enhance calcification and survival rates (Goreau, 2022; Hilbertz, 1979). Corals have shown enhanced photosynthetic efficiency (Huang et al., 2020) and growth under similar conditions (Strömberg, Lundälv & Goreau, 2010; Chavanich et al., 2013). Similar positive effects were also observed in oysters, which revealed a faster growth under low-voltage currents (Berger et al., 2012; Shorr et al., 2012). These studies documented the stimulation effect, while others support our findings on the limited effects. Chavanich et al. (2013) reported no significant benefits in coral growth under electric current conditions (2 A m−2 and 4 A m−2). Similarly, Strömberg, Lundälv & Goreau (2010) found no significant growth enhancement in scleractinian corals when exposed to various electric current densities (0.00–2.19 A m−2). Additionally, Piazza et al. (2009) revealed no notable increase in oyster growth during their experiment at low currents (ca. 2 A m−2). On the other hand, several studies have shown that the electrochemical deposition of CaCO3 strongly stimulates coral and oyster growth with growth rates reported to be 3–5 times faster than controls, and in some cases, reaching up to 6–20 times (Sabater & Yap, 2004; Goreau & Hilbertz, 2012; Goreau, 2012; Bakti et al., 2012; Fitri & Rachman, 2012; Jompa et al., 2012; Zamani, Abdallah & Subhan, 2012; Natasasmita, Wijayanti & Suryono (2016), 2016; Munandar et al., 2018). A summary of the study on the effects of the Biorock method using electric current to promote growth in corals and oysters is reported in Table 3. Building on these findings, it is important to consider the potential influence of host-specific factors on the observed responses.

Table 3 Summary of the study on the effects of the Biorock method using electric current in hermatypic corals and oysters.

Organism	Current density	Effects	Reference	
Benthic foraminifera (Amphistegina lobifera)	0.29–2.86 μ A/cm2	Positive but no significant growth enhancement	This study	
Coral
(Pocillopora damicornis)	2 A m−2, 4 A m−2	No growth enhancement	Chavanich et al. (2013)	
Coral
(Lophelia pertusa)	0.00–2.19 A m−2	No growth enhancement	Strömberg, Lundälv & Goreau (2010)	
Oyster
(Crassostrea virginica)	2 A m−2	No growth enhancement	Piazza et al. (2009)	
Oyster
(Crassostrea virginica)	6–9 V	2.7 times growth enhancement	Berger et al. (2012)	
Oyster
(Crassostrea virginica)	7.39 A, 11.15 A	9.30 times growth enhancement	Shorr et al. (2012)	
Coral
(Galaxea fascicularis)	10 mA m−2,
100 mA m−2	Growth enhancement at low current	Huang et al. (2020)	
Gorgonian soft coral
(Isis hippuris)	5 A	2.8 times growth enhancement	Fitri & Rachman (2012)	
Coral
(Porites cylindrica)	4.18 A	Growth enhancement	Sabater & Yap (2004)	
Coral
(Acropora tenuis and Acropora cytherea)	6 A	1.6–3 times growth enhancement	Zamani, Abdallah & Subhan (2012)	
Coral
(Acropora formosa)	1.2 V	3–4 times growth enhancement	Bakti et al. (2012)	
Coral
(Acropora nobilis)	6–12 V	4 times growth enhancement	Jompa et al. (2012)	
Coral (Acropora  micropthalma and Pocillopora  verrucosa)	6 V	4 times growth enhancement	Munandar et al. (2018)	
Coral (Acropora cerealis )	6–12 V	1.5 times growth enhancement	Natasasmita, Wijayanti & Suryono (2016)	
Coral (Porites porites and Acropora cervicornis)	6–12 V	6 times growth in P. porites
8 times growth in A. cervicornis	Goreau & Hilbertz (2012)	

Benthic foraminifera and other calcifying organisms, such as corals both contribute to carbonate production and sediment formation (Langer, Silk & Lipps, 1997). However, LBF account for a smaller portion with ∼43 million tons (Langer, Silk & Lipps, 1997), versus ∼900 million tons for corals (Milliman, 1993). However, their structural and physiological features are markedly different, with distinct metabolic requirements, calcification mechanisms, and responses to external stimuli, including electric currents. Indeed, it would also be stressed that LBFs, corals and oysters have mineralogical and calcification mechanisms differences. LBFs and other benthic foraminifera secrete high-magnesium calcite (e.g., Bentov, Brownlee & Erez, 2009), corals produce aragonite skeletons (McCulloch et al., 2017), while oysters primarily form calcite shells, occasionally producing aragonite in early development stages (Marin, Roy & Marie, 2012). Regarding calcification processes, in LBFs, calcification occurs extracellularly on the shell surface as new chambers are sequentially added. The process involves seawater vacuolization, transmembrane ion-transport, involvement of organic matrices and/or pH regulation (De Nooijer et al., 2014). In corals, calcification arises within the subcalicoblastic space, an extracellular compartment between tissue and skeleton, where the process is actively regulated through pH elevation and ion transport to facilitate skeleton formation (McCulloch et al., 2017). In both groups, photosynthetic symbionts play a key role in regulating internal carbonate chemistry to promote calcification (Bentov, Brownlee & Erez, 2009; McCulloch et al., 2017). Oysters, which lack symbionts, rely on a combination of active ion transport and passive diffusion, with shell formation controlled by organic matrices (Marin, Roy & Marie, 2012). These differences likely account for the contrasting outcomes observed in Amphistegina lobifera compared to other calcifying organisms like corals when exposed to electric current stimulation, highlighting the complex interplay between host physiology and symbiont activity.

Culturing protocols, technological improvements and further suggestions

Optimising experimental approaches is essential to enhance the growth of benthic foraminifera under controlled conditions. Advanced culturing protocols may be required to achieve notable growth enhancement as demonstrated at 1 and 1.43 µA/cm2 treatments in this study. The electrical current density plays a critical role in favouring the growth rate, with certain values being more effective for specific species. The electrical current density used in this study was lower compared with that used in the Biorock system and, therefore, may have resulted in a slower precipitation rate of new calcite. We used a lower current density (0.29–2.86 µA/cm2) that allows the survival of most individuals, as reported in the previous study on viability under electrical exposure (Rebecchi et al., 2023). Hilbertz & Goreau (1996) suggested that an optimal current density for promoting growth and mineral accretion lies within the range of 0.1–30 A/m2. Previous studies using a higher range have shown significant growth and mineral deposition in marine organisms indicating that stronger current density may yield more pronounced effects (Jompa et al., 2012; Karissa et al., 2012; Munandar et al., 2018). The exposure period is also a crucial factor. In this study, 30 days were chosen as an optimal time frame to allow for repeatability of the experiments but also allow for a substantial amount of newly built chambers. However, with a slightly different experimental design and duration, it may have been possible to observe significant growth effects, particularly around the 1.43 µA/cm2 treatment. Several studies on corals have reported that longer exposure times to electric current promote higher growth rates (Bakti et al., 2012; Goreau & Hilbertz, 2012; Sabater & Yap, 2004). The effects of electric current on the growth rate should also be tested on additional symbiont-bearing species such as A. lessonii, Heterostegina depressa, and Marginopora vertebralis. This is particularly important considering that A. lobifera is regarded as an invasive species in the Eastern Mediterranean (Raposo et al., 2023). These additional species share ecological traits and habitats with A. lobifera. Amphistegina lessonii co-occurs with A. lobifera in shallow, warm reef environments of the Red Sea and to lesser extent in the Mediterranean (Langer & Hottinger, 2000; Hyams, Almogi-Labin & Benjamini, 2002; Weinmann et al., 2013; Guastella et al., 2019). Heterostegina depressa and M. vertebralis are typical coral reef LBF of the Indo-Pacific and Red Sea and the former species is also reported in the eastern Mediterranean (Ross, 1972; Langer & Hottinger, 2000; Langer, 2008b). All three species possess calcareous tests, host photosynthetic symbionts, and have been used in experimental studies on LBF responses to climate change stressors (Uthicke & Fabricius, 2012; Schmidt, Kucera & Uthicke, 2014; Prazeres, Uthicke & Pandolfi, 2015; Reymond, Patel & Uthicke, 2022; Dämmer et al., 2023). Therefore, investigating the response of native and invasive species could help better understand species-specific responses and assess whether electrical stimulation might unintentionally favour the proliferation of opportunistic and invasive species and evaluate the ecological implications of applying such methods in different environmental settings. Moreover, juvenile foraminifera, given their smaller size, are potentially more responsive during early developmental stages when exposed to electric currents, however, results may also be relative to higher overall growth in the control and not lead to overall stronger effects. Furthermore, increasing the number of specimens used in the experiments would have provided a stronger statistical dataset, allowing for more reliable trends regarding the effects of electric currents on foraminiferal growth. Environmental factors, such as nutrient water concentration, which were not the primary focus of this study, could have also influenced the results.

The type and dimensions of the electrodes are also pivotal for optimizing electrochemical processes. In this study, steel electrodes were used for both the cathode and anode, which were smaller in size compared to those used in the Biorock system setup. In an earlier study, Rebecchi et al. (2023) used platinum for both electrodes for similar applications, utilizing a smaller experimental setup with a six-well plate. The Biorock system suggests using electrically conductive materials for the cathode, which are protected from corrosion by the applied electrical current, and smaller and non-corrodible, non-toxic materials like coated titanium for the anode, which provide optimal results (Goreau, 2012). Future research will explore alternative materials for the cathode, such as carbon fiber, and potentially larger platinum electrodes as the anode to improve system performance.

Although this size increase was not statistically significant, the trend provides a basis for further investigation to unravel the effects of electric stimulation on foraminiferal growth. These initial experiments within this narrow stimulation range suggest that even low levels of electric stimulation can influence growth in LBF. Future studies should also assess other factors that may enhance growth, beyond electric stimuli. Considering the above mentioned culturing and technological improvements along with the previous experiments based on coral and other marine organisms, this approach can be applied in coral reefs where LBF thrive to potentially stimulate the recovery and the restoration of such sensitive ecosystems. However, for benthic foraminifera, several aspects remain unknown, including optimal current intensity, duration, long-term benefits, and energy requirements, which should be considered before any potential field application. Electrical stimulation has shown potential to tackle the effects of extreme stresses on corals such as bleaching (Goreau, 2022). The same author reported the overgrowth of corals and other marine organisms (e.g., mussel, tunicate, oyster among others) as well as entire reefs spontaneously flourishing all around Biorock structures.While these results cannot yet be directly applied to benthic foraminifera, they indicate potential directions for future research.

Conclusions

This study represents the first direct evaluation of the effects of different electric current densities on the growth rate and photosynthetic activity of the LBF A. lobifera over a 30-day period. Here, we provide evidence that the applied electrical current (0.29_2.86 µA/cm2) stimulation has no negative impact on the photosynthetic activity of its diatom symbionts as revealed by the maximum quantum yield (Fv:Fm) and Chl a content. Additionally, we observe a slight increase in growth rate, as supported by the formation of new chambers marked by the calcein label. While our study contributes to the growing body of literature on the effects of electric current on marine organisms, it highlights the need for further comparative studies involving different symbiont-bearing foraminiferal species and also different types of marine species. Increased chamber formation at 1 and 1.43 µA/cm2 treatments suggests that while electric current may influence the growth of benthic foraminifera, the conditions necessary for significant enhancement remain to be investigated. Future research should focus on optimising experimental conditions and exploring the potential of electric stimulation to promote growth in benthic foraminifera, providing insights into potential applications in marine biotechnology.

The authors are very grateful to the reviewers for their thoughtful and valuable comments that have greatly improved our contribution. We are grateful for the support of the technicians Stefanie Bröhl and Epiphane Yéyi for supporting experiments at the MAREE facility (ZMT, Centre for Tropical Marine Research). We thank the working group of Prof. Michal Kucera (MARUM, University of Bremen) for access to the PAM fluorometer.

Additional Information and Declarations

Competing Interests

Author Contributions

Data Availability

The authors declare there are no competing interests.

Federica Rebecchi conceived and designed the experiments, performed the experiments, analyzed the data, prepared figures and/or tables, authored or reviewed drafts of the article, and approved the final draft.

Davide Lattanzi analyzed the data, authored or reviewed drafts of the article, and approved the final draft.

Sigal Abramovich analyzed the data, authored or reviewed drafts of the article, and approved the final draft.

Patrizia Ambrogini analyzed the data, authored or reviewed drafts of the article, and approved the final draft.

Fabrizio Frontalini conceived and designed the experiments, analyzed the data, prepared figures and/or tables, authored or reviewed drafts of the article, and approved the final draft.

Christiane Schmidt conceived and designed the experiments, performed the experiments, authored or reviewed drafts of the article, and approved the final draft.

The following information was supplied regarding data availability:

The data is available at PANGEA: Rebecchi, Federica; Lattanzi, Davide; Abramovich, Sigal; Ambrogini, Patrizia; Frontalini, Fabrizio; Schmidt, Christiane (2025): Effects of electric current on growth, chlorophyll a content, pulse amplitude modulation response, and new chamber formation of the foraminifera Amphistegina lobifera [dataset bundled publication]. PANGAEA, https://doi.org/10.1594/PANGAEA.983600.

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
