# Peer review of "Effects of low-level electric current on the growth of Amphistegina lobifera and its photosynthetic diatom endosymbionts"

_PeerJ, doi:10.7717/peerj.20160_

## Round 0.1 · original submission · Major Revisions

Reviewer 1 ·

Basic reporting

The language of the manuscript is very clear.
Citations are sufficient and up-to-date.
A general context to justify their study is provided and well-explained. Yet there is still some room for improvement.
Raw data is shared, but the Data Availability statement is missing - will it be publicly shared? The experimental design is visualised in Fig. 1; results are represented clearly.
The advantages and limitations are explained; the study does not oversell itself, and it aims to improve our understanding of the application of available methods on different marine organisms (in this case, benthic foraminifera) and to expand possibilities.

Experimental design

Experimental design seems to be sufficient. Although the approach is not novel, given that the method has been applied on coral reefs to stimulate the growth of the corals and previously by the most of the co-authors of this study on a different benthic foraminifera species under a different controlled experimental setting, it fits well with the aims and scope of the journal and can be considered relatively original as it aims to observe the growth rate of A. lobifera under different electric currents.

The research question is obvious. The authors aim to understand if they can stimulate and accelerate the growth of large benthic foraminifers under the climate change scenarios to compensate for the carbonate dissolution related to increased pCO2. They perform a well-designed experiment in controlled conditions using living benthic foraminifera collected from the Eastern Mediterranean Sea. The methods are explained clearly, making it possible to replicate.

One week point is, the authors did not give a concrete reason for their choice of species, given that A. lobifera is an invasive species that reaches the Mediterranean waters from the Red Sea via the Gulf of Suez, I do not see any reasonable explanation why this particular species should be considered to help sustain the coral reefs but not another symbiont-bearing species such as Amphistegina lessonii.

Validity of the findings

Findings are meaningful, replicable. Their results are robust and statistically sound. The data - I do not see any Data Availability statement. I do not know if the authors will keep the data as a supplementary file. I would recommend making the data publicly available following FAIR principles.

Conclusions are mainly linked to the supporting results, except for the statement on the species-specific nature of their findings. The authors do not compare two or more different species to make such a statement. They suggest in their experiment that the electric current did not stimulate the photosynthetic activity it had no impact on foraminifera growth. Referring to the findings of Goeyse et al (2021), who conducted a completely different experiment on the effect of photosynthesis on A. lessonii growth, that inhibiting photosynthesis reduces calcification, suggesting that photosynthesis actively promotes calcification, they make a statement on the species-specific nature. I find this a bit far-fetched, and I would suggest avoiding such a decisive statement, but using this information for further experiments on different species.

Additional comments

lines 426-427: The authors mention species-specific nature; their study is based on one species only. I do not see any evidence in their research to support this statement. And indeed, this is one weak part in this study. Even though one might consider it preliminary research that has room to improve, A. lobifera is not the only species associated with coral reefs. True that it thrives in Indo-Pacific and Red Sea waters, but it is an invasive species in the Eastern Mediterranean and Aegean waters, migrating from the Red Sea. I cannot help but wonder, if this species were to grow faster to avoid the effects of global warming and pCO2 increase, what would be the aftermath on its dominance in the Red Sea and eventually in the Eastern Mediterranean? It is true that if this method has an impact on the reproduction of marine organisms is poorly studied (Goreau, 2022), but nonetheless, a better approach might be focusing on a different species that is not invasive. I believe the authors should discuss this issue that they have overlooked in their manuscript.

·

Basic reporting

In their submitted manuscript entitled “Effects of low-level electric current on the growth of Amphistegina lobifera and its photosynthetic diatom endosymbionts”, the authors present new data from a culture experiment of Amphistegina, a large benthic foraminifera, as a function of an electric current applied between plates immerged in the jars in which the individuals grow. The authors reports Chla, growth rates and number of new chambers and interpret the data in terms of how the applied current influences the photosynthetic activity and thus the growth rates.
They found no significant variations as a function of the applied current and compare their results to other similar experiments. As a negative result is a result, the authors interpret their results in order to explain why in LBF there is no significant influence of the electric current unlike what can be seen in corals and oysters.

However the flow of the article could probably be improved in a way that would make the contribution more straightforward, and of greater interest.
In addition, a few major points need to be addressed (detailed in sections 1, 2 and 3).

One of the main points to adress is that research question would benefit to be more clearly defined: is the goal to assess the best material for reef reconstruction so that it does not harm LBF? understand the influence of electric current on biocalcification of LBF? On photosymbionts? Below are a few reasons why a clearer research question would help:

1/Even though the authors work on forams, they start their abstract and introduction with coral reefs. It seems more logical to have the introduction start with forams, and how acidification/changes in alkalinity impacts them, and why the electric current method is relevant or of interest to them, as their growth mechanisms and locations are not the same of those of corals.

2/ The introduction states that the electrode works by accelerating calcification rate (line 61) because of an increase of HO- ions. Overall, the influence of pH is mentioned as key in the introduction and never really discussed again. How does seawater chemistry evolve in the course of the experiments?

3/The influence of electric currents on the symbionts is not mentioned in the introduction. Yet,line 308 and 329, the authors state that “Contrary to our hypothesis, the electric current has not stimulated the photosynthetic activity”. However, no such hypothesis has been formulated in the introduction. Rather, the introduction leaves the reader with the impression that the hypothesis is that the electric current increases HO- production, thereby increasing pH, thereby increasing saturation state and thus carbonate precipitation.

4/The discussion starts with the symbionts, and a long paragraph that starts line 285 provides those elements. This paragraph should probably be moved to the introduction. This is also true from the paragraph about CA (Line308-326). The first section of the discussion is entitled “Effects […] on […]endosymbionts” but starts with growth rates. Maybe here as well a reorganization would help.

5/ CA seems to be treated as one enzyme, when there are actually many different types of CAs. This could be discussed. Could some of them be directly impacted by an electric current? Could it impact the structure of the enzyme? It is important to underline that CA does accelerate the interconversion between CO2 and HCO3-, there making it a non-limiting step in DIC equilibria. In many organisms, instead of making more CO2, CA is considered as a step that allows on the contrary CO2 to turn into HCO3- and eventually CO32- (e.g. Chen et al., 2018, GCA; Pesnin et al., 202, Chemical Geology). This only appears later in the text (line 320 – 325).

6/Finally, ending the discussion on experimental set-up, which is barely discussed in the introduction, does not seem like the best way to finish. Maybe for information could be provided on the topic in the introduction. The presentation about CA, like that of symbionts, and of materials used as plates, should all move up to the introduction to help the authors state their working hypothesis.

Minor corrections : line 198 diamenter instead of diameter
Line 258: a higher percentage […]was labelled
Line 261 ± symbol missing
In figure 2, what is the error used by the authors (standard error I assume)?

Experimental design

1/ As the authors mention, some of the methods are not relevant (line 300-305). As a lot of the discussion concerns the influence of the electric stimulation on the symbionts, which is described line 121 as one of the goals of the study. As a result it would be useful to rely on the most accurate method to estimate symbiont activity.

2/ Some points are not discussed: is a system applied to coral reefs transposable to environments where the LBF grow? What would be the interest? The authors remind that Langer estimated LBF at 80% of foraminferal reefal contribution, but this represents only2.5% of global carbonate production, and LBF are much more diffuse than corals. Coral reef restauration has been presented has a way to fight against beach erosion, for instance. What would be the end goal with LBF? Regarding the distribution of LBF, what could be the implication of applying a weak current in natural ecosystems on a broad scale? How much H2/HO- is generated?

3/ Differences between corals, oysters and LBF are not discussed in terms in mineralogy (calcite vs. aragonite) or biomineralization processes (e.g.Passive vs. active transport, diffusion, alkalinity pumps, vacuoles). In addition, a comparison is necessary to anything that is known about other experiments where alkalinity/pH is raised in (benthic) foraminifera growth experiments. How can the authors tease apart the influence of the current itself to that of seawater chemistry/H2 concentrations?

Validity of the findings

1/ A first major point is that pH values as well as seawater composition, salinity and temperature should be provided.

2/Regarding the data reporting, is the number of significant figures accurate for size measurements ? There are more figures for the size measured at the beginning than at the end of the experiments, and no errors. Errors should be provided, as error on the measurements will have consequences on the growth rates estimates.

3/ The raw data come from three different jars. I would suggest the figures to represent each set of 16 specimens represented with their own value and standard deviation. Also, I recalculated the mean growth rates values for each current and did not find the same values as the authors. Line 275 : the one highest growth rate is indeed observed at 1.43 µA/cm2, but it’s not the highest average (at least in what I recalculated). And the result is interpreted at this point in terms of calcification/chamber formationI report here my calculation so that the authors can double check what they provided against what I’ve calculated from the available data, with the standard deviations depending on the population used, the mean of the jars, of the mean of all forams :


µA/cm2 jar Mean +- Stdev(n=16) Mean exp+-stdev (n=3) Mean on all values (n=48)

0.00 CT2 2.59+-2.32 3.25 +-0.84 3.25 +-2.52
0.00 CT3 2.97+-2.49
0.00 CT1 4.19+-2.62
0.29 A3 1.74+-2.55 3.09 +-1.86 3.09 +-3.03
0.29 A2 2.31+-2.03
0.29 A1 5.21+-3.28
1.00 B1 2.10+-1.57 3.65 +-1.36 3.65 +-3.60
1.00 B2 4.19+-4.36
1.00 B3 4.66+-3.90
1.43 C2 3.33+-2.21 3.60 +-0.26 3.65 +-2.54
1.43 C1 3.61+-3.54
1.43 C3 3.85+-1.65
2.86 D2 2.07+-1.71 3.00 +-0.93 2.98+-2.64
2.86 D1 3.00+-2.80
2.86 D3 3.92+-3.60

I do however find the same overall values for the number of chambers. Is the difference because the authors removed all zero/negative values from the growth rates values? If they did so, why? One specimen seems to have disappeared from the highest current experiment (-100%), and I did not see where this was commented. It should be, as the value seems to have also been removed from the final calculations.

4/ I am not an expert in statistics, and I trust the tests performed by the authors. However, plotting their data with average value and standard deviations as estimates of the variaibility for Chla or growth rates seems to reveal no significant variations. Using standard error has, in my understanding, the underlying assumption that all samples should have the same value and it provides an error on the estimate of the mean. As a result, I would use standard errors for successive measurements of the growth rate for a single foram but not, as here, to document variability within a natural/global population.Instead, I would rather use standard deviations.

5/. Line 331 : the authors comment on a peak that they describe as “non-significant”. This peak at 1.43 µA/cm2 is central in a lot of their discussion and I did not find it in my recalculation of the growth rates. In addition, the peak is not significant in the authors' calculations. The authors might probably need to rethink their discussion and conclusion in this regard.

Finally, are new chambers reported as a number of new chambers or a rate? There seems to be contradictions between the reported data (“number of new chambers”) and most of the text(e.g. line 264 or 276 “% newly formed chambers”: how is this value calculated? What is the error on it?) and figures. In addition, is the number of significant figures relevant?
All values used could be presented in a table at some point.

Additional comments

I thank the authors and the editor for the opportunity to review this work and I hope my comments will be of use.

Best regards,

Guillaume Paris

---

## Round 0.2 · Minor Revisions

We have received evaluations from two expert reviewers. Both comment that this version has much improved, however they both comment that there are still minor issues that need to be dealt with prior to acceptance.

Please ensure that you address all of these suggestions in a revised version and clearly show where modifications have been made to the manuscript.

Reviewer 1 ·

Basic reporting

The authors have made a significant improvement in their manuscript and addressed the issues reviewers raised. I appreciated that they also mentioned A. lobifera being an invasive species, and similar experiments should concentrate on other species too, to understand potential unintentional consequences that this methodology might result in.

The language of the manuscript is clear and informative. Cited references are sufficient. Data is shared, and figures are clear. Results are relevant.

I suggest adding references for the suggested species to expand this particular experiment on, i.e., line 410, A. lessonii, Heterostegina depressa, and Marginopora vertebralis, and the rationale behind this choice to demonstrate that this statement is not an afterthought. Do they live in the same environment as A. lobifera? Are they common in the Mediterranean Sea or the Red Sea?

Experimental design

The study is original, the experiment is adequate, the research question is well-defined, and it aims to fill a knowledge gap of using electrochemical stimulation to promote the precipitation of calcium carbonate as a bioremediation method in sensitive carbonate environments such as coral reefs, targeting an important group of carbonate producers (foraminifers). Methods are described in detail and are hence replicable.

Validity of the findings

Findings are important and open the door to new research questions. The data is provided, statistically assessed. Conclusions are linked to the research question.

Additional comments

line 88: CO₃² ---> CO₃²⁻
line 94: (Langer et al., 1997, Langer 2008) ---> (Langer et al., 1997; Langer 2008)
line 167: climate control room ---> climate-controlled room
line 414: help to better understand ---> help better understand

·

Basic reporting

Dear authors, please find below my new review of the draft submitted to PeerJ.

I thank the authors for this new version, which clarifies and provides useful modifications. I do believe, however, that some points still need additional explanations.

The authors have clarified their working hypothesis. They now state line 109. Similar to studies in corals, based on this physiological interplay, we hypothesize that electrical stimulation might enhance photosynthetic activity and, thereby, promote growth and calcification in LBF. I thank the authors for making their hypothesis clearer. However, they should probably state in the text that at this point, the direct effect of pH increase due to the electrical current is not considered and that interpretations based on the effect of the electric current thus have to leave aside the effect of ambient pH increase, as they explained in their rebuttal.

The new version of the article is well written, though a possible last round will help correct some typos and fluidize some sentences. Below are some examples :

Line 64: when supersaturation is reached
Line 96: due to their abundance
Line 368: "Benthic foraminifera and other calcifying organisms, such as corals, are both significant contributors to carbonate production and sediment formation (Langer et al., 1997)."
->I would suggest rephrasing this sentence to not leave the impression that they are equal contributors, as foraminifera in total represent only a subfraction of reef production (43 million tons for forams vs 900 million tons for corals), and that reef carbonates are only a fraction of the total calcareous production. Similarly, in the introduction, the authors should mention that LBF represents 80 % of foraminiferal reef production, which itself represents ~5% of the total reef production.

In the introduction, the authors should mention that the effect of CO2 increase is not the same on all organisms. Interestingly (and in contradiction with the text as written), de Goeyse et al. (2021) suggest that Amphistegina lessonii could benefit from the increase in DIC associated with the atmospheric pCO2 increase. The introduction thus needs to be more nuanced and should cite Goeyse et al.'s finding.

Line 104, when documenting pH increase at the SOC, Rollion Bard et al., 2010, should be cited for their work on Amphistegina lobifera; as well as Sutton et al. (2018) for their estimate of pH at the SOC of various organisms.

Finally, Line 104, the authors write, "In benthic foraminifera, calcification is linked to proton pumping, which increases the pCO¢ directly outside the site of calcification (SOC) by converting bicarbonate into CO2. This process creates a strong inward/outward pCO2 gradient, promoting CO2 diffusion into the cell." I’m not sure I understand the sentence as it is written. My understanding is that proton pumping increases alkalinity and pH at the SOC, which favors CO32- formation and sustains CO2 diffusion towards the SOC, and not into the cell as written. This is of importance, as line 335, the authors further write that “photosynthesis actively promotes calcification, potentially through CA's role in increasing CO2 availability outside the cell”. This only makes sense if the latter sentence should be read as “photosynthesis actively promotes calcification, potentially through CA's role in increasing CO2 availability at the site of calcification”. Please clarify.

Experimental design

-

Validity of the findings

The findings are valid, and the new presentation makes them clearer.

I have a comment. Line 325, the authors write that "Our study also shows that the use of the calcein probe is a useful and reliable method for identifying newly formed chambers". This appears to me like a routinely used approach, and it has been shown before (eg, Barras et al., 2009). Please remove, or rephrase and cite previous studies that showed the same result.

Additional comments

In this submitted draft, a point that is still missing is the environmental impact of the proposed method. No critical review of the method is proposed.

First, I'm curious to know if there are studies that show that it is necessary to restore reefs for tourists to mitigate climate change and ocean acidification.

More importantly, to restore a reef, how much current is necessary? And for how long does it need to be applied? Does the benefit of the process stay, or stops once the current is no longer applied? The energy consumption required should be mentioned to give an idea of the feasibility and environmental cost of the approach suggested.

Even if that is not the direct scope of the article, those points should be at least mentioned as either an unknown or a caveat. This is of importance as it would make little sense to suggest a remediation system that requires too much energy or could actually contribute to emitting more CO2, depending on the source of the current applied.

---

## Round 0.3 · accepted · Accept

Thank you for your detailed responses to the reviewer comments. I am satisfied that you have covered all of the issues raised and I am proposing that this manuscript be accepted.